# G3BP1 Regulates the Cell Cycle by Promoting IFNβ Production to Promote PCV2 Replication and Promotes Nuclear Transfer of Viral Proteins by Direct Binding

**DOI:** 10.3390/ijms26031083

**Published:** 2025-01-27

**Authors:** Xinming Zhang, Kang Li, Shenglan Zhou, Leyi Zhang, Lei Wang, Yanling Liu, Shuangyun Wang, Ge Xu, Pengshuai Liang, Zheng Xu, Changxu Song

**Affiliations:** College of Animal Science & National Engineering Center for Swine Breeding Industry, South China Agriculture University, Guangzhou 510642, China; xmzhang202211@163.com (X.Z.); 18058940715@163.com (K.L.); zhoushenglan_z@163.com (S.Z.); zhangleyi210@163.com (L.Z.); wangllei2024@163.com (L.W.); yll395070943@163.com (Y.L.); wsy547089123@163.com (S.W.); xuge970922@163.com (G.X.); lps257@163.com (P.L.)

**Keywords:** PCV2, G3BP1, IFNβ, cell cycle arrest, direct interaction

## Abstract

Porcine circovirus type 2 (PCV2) is a significant pathogen responsible for porcine circovirus-associated diseases (PCVAD), and it is widely prevalent in pig farms, leading to huge economic losses for the pig industry. Currently, the ability of PCV2 to enhance its own replication by using the antiviral inflammatory factors IFNα, IFNβ, and IL-2 and its complex immune escape mechanism remain unclear, which has attracted wide attention. Research has indicated that GTPase-activating protein (SH3 domain)-binding protein 1 (G3BP1) is involved in the innate immune response to a variety of viruses, primarily by regulating and composing stress granules (SGs) to inhibit viral replication. Our initial studies identified elevated G3BP1 expression during PCV2 infection, paradoxically promoting PCV2 replication. In light of this phenomenon, this study aims to elucidate how PCV2 regulates G3BP1 to enhance its replication. Our findings demonstrate that G3BP1 overexpression further activates PCV2-induced expression of RIG-I, MDA5, cGAS and STING, thereby promoting IFNβ production and affecting cell cycle arrest in the S phase, facilitating PCV2 replication. Moreover, interactions were observed between PCV2 Cap protein and G3BP1’s RGG domain, and between PCV2 Rep protein and G3BP1’s NTF2 and RRM domains, potentially promoting viral protein nuclear transfer. In summary, PCV2 enhances its replication by modulating G3BP1 to induce IFNβ production and directly binds viral proteins to promote viral protein nuclear transfer. This research provides a foundation for further investigation into the immune evasion mechanisms of PCV2.

## 1. Introduction

Porcine circovirus type 2 (PCV2) was initially identified in swine exhibiting Postweaning Multisystemic Wasting Syndrome (PMWS) in 1991 [1,2]. Subsequent research has established a correlation between PCV2 and various health issues, including necrotizing pneumonia, encephalopathy, diarrhea, dermatitis, kidney disease, congenital tremor, and reproductive disorders, which are now collectively termed porcine circovirus-associated diseases (PCVADs) = [3,4,5]. Furthermore, PCV2 infection has been shown to induce immunosuppression in pigs, leading to heightened vulnerability to other pathogens, diminished immune responses to vaccinations, and the potential for interspecies transmission. Consequently, PCV2 represents a significant threat to the global swine industry and public health, underscoring the urgent need for further investigation into the pathogenic mechanisms and immune evasion strategies employed by this virus.

Porcine circovirus type 2 (PCV2) is one of the smallest viruses discovered to date, with a particle diameter of 15–20 nm. It consists of a 60-unit capsid protein (Cap) forming an icosahedral structure enclosing single-stranded covalently closed circular DNA, also known as ssDNA. The ORF1 gene, located on the positive strand of the circular DNA, encodes the replication enzyme Rep and Rep’ proteins in a clockwise direction, primarily involved in the virus’s rolling circle replication process [4]. Extensive research indicates that PCV2 largely achieves host immune escape through interactions between its own proteins with various host cell factors. For example, PCV2 cap binds to DEAD-box RNA helicase 21 to promote viral replication [6]. In addition, PCV2 Cap inhibits PKR activation by interacting with Hsp40 [7]. It is worth noting that PCV2 infection increases the protein levels of phosphorylated IRF3 (p-IRF3), mitochondrial antiviral signaling protein (MAVS), retinoic acid-inducible gene I (RIG-I), and melanoma differentiation-associated protein 5 (MDA-5) [8]. The activation of these host innate immune molecules further promotes the expression of downstream interferon-stimulated genes (ISGs). This dysregulates intracellular inflammatory cytokine expression and disrupts cellular homeostasis, creating favorable conditions for the invasion of other diseases [2]. Interestingly, while type I interferons (IFN-α and IFN-β) and interleukin-2 (IL-2) are well-known for their antiviral effects, their stimulation paradoxically enhances the replication of PCV2 [8,9].

Ras-GTPase activating protein (SH3 domain)-binding protein 1 (G3BP1) is a critical protein involved in cellular stress responses, controlling and participating in stress granule assembly. It participates in various biological processes and disease pathogenesis, regulating mRNA metabolism, dynamically modulating stress granules, and influencing viral replication and development [10,11]. G3BP1 consists of five distinct domains: the nuclear transport factor 2 (NTF2) domain, RNA recognition motif (RRM), acidic region, proline-rich (PxxP) domain, and arginine-glycine-glycine (RGG) domain. The NTF2 domain is crucial for viral replication as it binds to viral motifs and is recruited by viral replication complexes (Yang et al., 2020) [12]. The RGG domain, enriched in arginine and glycine residues, enhances the RNA-binding affinity of the RRM domain and mediates protein–protein interactions, which are essential for recruiting the host translation machinery. The complex structure of G3BP1 dictates its diverse functions. As an antiviral component, G3BP1 promotes the signaling of retinoic acid-inducible gene I (RIG-I)-like helicases (RLH) by upregulating the expression of RIG-I and melanoma differentiation-associated gene 5 (MDA5), thereby enhancing the interferon-β response upon binding viral dsRNA and RIG-I [13]. Furthermore, G3BP1 facilitates the binding of cyclic GMP-AMP synthase (cGAS) to DNA, activating the synthesis of 2′,3′-cyclic GMP-AMP (cGAMP). cGAMP binds to endoplasmic reticulum proteins, further activating downstream pathways and inducing the production of pro-inflammatory cytokines [10]. Many viruses exploit G3BP1 as part of their immune evasion strategies. For instance, the N protein of porcine reproductive and respiratory syndrome virus (PRRSV) interacts with G3BP1, leading to G3BP1 phosphorylation and impairing its antiviral function [14]. Porcine epidemic diarrhea virus (PEDV) infection induces caspase-8-mediated cleavage of G3BP1, disrupting stress granules and promoting viral replication [15]. Interestingly, our previous research found that porcine circovirus type 2 (PCV2) infection enhances G3BP1 expression, However, the overexpression of G3BP1 paradoxically promotes PCV2 replication. The underlying mechanisms of this phenomenon remain unclear.

This study aims to investigate the reciprocal regulation between G3BP1 and PCV2, providing novel insights into the mechanisms of interaction between G3BP1 and DNA viruses. Additionally, this research establishes a foundation for further exploration of the pathogenic mechanisms and immune evasion strategies employed by the PCV2 virus.

## 2. Results

### 2.1. PCV2 Infection Significantly Promotes the Expression of G3BP1

As illustrated in Figure 1A,B, the infection of PK-15 cells with the PCV2 virus (MOI = 1) significantly enhanced the expression of G3BP1 mRNA and protein at 12 and 24 h post-infection. To further investigate whether PCV2 increases G3BP1 expression through its viral proteins, Cap and Rep, we transfected PK-15 cells with plasmids encoding these proteins. The results demonstrate that both PCV2 Cap and Rep proteins significantly upregulated G3BP1 expression at both the mRNA and protein levels, with the effect of Rep protein being more pronounced (Figure 1C,D). Additionally, to determine whether PCV2 infection also promotes G3BP1 expression in vivo, we infected porcine primary alveolar macrophages (PAM) with the PCV2 virus at an MOI of 1. Our analysis revealed that PCV2 continued to enhance G3BP1 expression (Figure 1E,F). Similarly, transfection with the viral proteins also significantly increased G3BP1 expression (Figure 1G,H).

### 2.2. Overexpression of G3BP1 Promotes PCV2 Replication

Previous studies have demonstrated that cells infected with PCV2 can significantly enhance the expression of G3BP1. To further investigate the role of G3BP1 protein in viral infection, we overexpressed G3BP1 in PK15 cells infected with PCV2, and the results are shown in Figure 2A–D; compared to the control group, both relative expression levels of Cap and Rep and virus copy numbers, as well as protein levels, indicate a promoting effect of G3BP1 on PCV2 replication. This suggests that PCV2 facilitates its own replication by upregulating the expression of G3BP1.

### 2.3. Silencing the Expression of G3BP1 Inhibited the Replication of PCV2

To assess the impact of G3BP1 interference on PCV2 replication, we designed three siRNAs. Subsequently, we verified the silencing efficiency of these siRNAs. As illustrated in Figure 3A,B, SiG3BP1-1 and SiG3BP1-3 exhibited effective interference at both the gene and protein levels, achieving silencing efficiencies exceeding 75%. For subsequent experiments, we selected a 1:1 mixture of SiG3BP1-1 and SiG3BP1-3 for transfection to inhibit G3BP1 expression in PK-15 cells. Following this, we silenced G3BP1 and observed the effects of continuous infection on PCV2 Cap and Rep mRNA expression levels, viral copy number, and protein expression, as depicted in Figure 3C–F. The results indicate that reducing G3BP1 expression significantly inhibits PCV2 replication.

### 2.4. G3BP1 Promotes PCV2-Induced Expression of RIG-I, MDA5, MAVS, cGAS, and STING

Research has demonstrated that PCV2 infection of PK-15 cells activates both the RIG-I/MDA5 signaling pathway and the cGAS/STING signaling pathway, promoting the production of IFNβ and thereby facilitating PCV2 replication [8,9]. G3BP1 also plays a crucial role in regulating the RIG-I/MDA5 and cGAS/STING signaling pathways. To further verify whether G3BP1 promotes RIG-I and MDA5 transcription induced by PCV2-infected PK15 cells, we overexpressed G3BP1 in PK15 cells infected with PCV2. Detecting the mRNA levels of RIG-I and MDA5, the results are shown in Figure 4A,B, we found a significant promotion of RIG-I and MDA5 transcription compared to the control group upon overexpression of G3BP1. Similarly, overexpression of G3BP1 may further enhance the activation of the PCV2-induced expression of cGAS and STING. As shown in Figure 4C,D, PCV2 infection promotes the expression of cGAS and STING mRNA, with the overexpression of G3BP1 further enhancing the transcription of PCV2-induced cGAS and STING mRNA.

### 2.5. G3BP1 Promotes PCV2 Replication by Triggering IFNβ Expression and Arresting Regulating Cell Cycle at S Stage

G3BP1 upregulates the expression of RIG-I, MDA5, MAVS, cGAS, and STING induced by PCV2, thereby triggering more intense inflammatory responses and promoting the release of inflammatory factors. Dual fluorescence reporter gene assays conducted in PK15 cells infected with PCV2 demonstrated that the overexpression of G3BP1 significantly enhances IFNβ promoter transcription (Figure 5A). Real-time fluorescence quantitative detection demonstrated that the overexpression of G3BP1 enhanced PCV2-induced IFNβ expression in PK-15 cells infected with an MOI = 1, with the most significant enhancement observed at 12 h compared to the control group (Figure 5B). To further confirm the enhancing effect of IFNβ on PCV2, PK-15 cells were stimulated with 500 ng/mL porcine IFNβ, and the impact on PCV2 Cap and Rep mRNA expression levels, as well as viral gene copy numbers, at different time points was assessed (Figure 5C–E). The results indicate that IFNβ treatment significantly promotes PCV2 replication, with the most pronounced effects observed at 36 and 48 h.

Multiple studies, including this study, have confirmed the phenomenon of IFNβ promoting PCV2 replication. To further investigate the underlying mechanisms, we conducted a transcriptomic analysis of PCV2-infected groups treated with IFNβ and normal PCV2-infected groups. Our analysis revealed a significant enrichment of cell cycle signaling pathways in the IFNβ-treated PCV2-infected group, accompanied by a marked decrease in the expression of Cyclin A (Cyc A), a key protein that regulates the S phase of the cell cycle [16]. Previous research has indicated that PCV2 replication necessitates the host’s DNA polymerase to complete its rolling-circle replication, a process that typically occurs during the S phase of the host cell cycle, specifically during DNA synthesis phase [17]. Therefore, IFNβ may facilitate PCV2 rolling-circle replication during the S phase by regulating the cell cycle. Subsequently, we validated the regulatory effects of PCV2 and IFNβ on the Cyc A gene, demonstrating that both PCV2 infection and IFNβ stimulation inhibit Cyc A gene expression. This inhibition exhibits a synergistic effect, resulting in a more pronounced reduction in Cyc A transcription levels (Figure 5F). Furthermore, we assessed the impacts of PCV2 infection, IFNβ treatment, and their combined action on the cell cycle. The results reveal that IFNβ stimulation induces cell cycle arrest at the S phase in PK15 cells, and when combined with PCV2 infection, nearly all cells are arrested in the S phase (Figure 6A,B). These findings indicate that IFNβ can suppress Cyc A gene expression and induce cell cycle arrest at the S phase. Finally, Cyc A gene silencing was performed with an efficiency exceeding 60% (Figure 6C). As shown in Figure 6D–F, following Cyc A gene silencing, the expression levels of PCV2 Cap and Rep mRNA and viral gene copy numbers demonstrated significantly enhanced PCV2 replication. To further verify whether the increased PCV2 replication was related to the duration of cell cycle residence in the S phase, we treated PK-15 cells for 2, 4, 6, 12, and 24 h with the reversible DNA synthesis inhibitor thymidine riboside (TdR) to prolong the duration of cell residence in the S phase. Concurrently, PCV2 virus with MOI = 1 was inoculated and the difference in virus copy number was observed after a 24 h culture period. The results are shown in Figure 6G. With the increasing duration of TdR treatment time on PK-15 cells, the virus copy number increased significantly. This indicates that the enhancement of PCV2 replication is closely related to the length of time that the cell remains in the S phase of the cell cycle.

### 2.6. PCV2 Cap and Rep Proteins Interact with G3BP1

Previous studies have demonstrated that the Cap protein of PCV3 can interact with G3BP1 [18]. Given that PCV2 Cap and Rep proteins also regulate G3BP1, it is essential to further investigate whether a direct interaction exists between them. We co-transfected HEK293T cells with plasmids expressing PCV2 Cap, Rep, and G3BP1 proteins, and observed their co-localization using confocal microscopy. As illustrated in Figure 7A,B, PCV2 Cap and Rep proteins indeed co-localized with G3BP1 protein. Both PCV2 Cap and Rep proteins possess nuclear localization signals and are typically localized in the nucleus [19]. However, the images reveal that G3BP1 protein co-localized with PCV2 Cap and Rep proteins in the cytoplasm, indicating a strong affinity. Subsequently, we performed immunoprecipitation to further confirm the interaction between PCV2 Cap, Rep proteins, and G3BP1 (Figure 7C,D). To identify the functional regions involved in the interaction between PCV2 Cap, Rep proteins, and G3BP1, we truncated them based on nuclear transport factor 2 (NTF2) domain, acidic region, proline-rich (PxxP) domain, RNA recognition motif (RRM), and arginine-glycine-glycine (RGG) domain. The immunoprecipitation results demonstrate that G3BP1 protein requires the RGG domain to bind with Cap protein, while the NTF2 and RRM domains are necessary for binding with Rep protein. Thus, PCV2 Cap interacts with the RGG domain of G3BP1, whereas PCV2 Rep interacts with the NTF2 and RRM domains of G3BP1 (Figure 7E,F). It is known that PCV2 assembles completely in the nucleus, and the transfer of mature viruses from the nucleus may require the involvement of host proteins. Given the interaction of PCV2 Cap and Rep proteins with G3BP1 in the cytoplasm, we hypothesized that G3BP1 may be involved in the nuclear translocation following PCV2 maturation. We overexpressed G3BP1 in PK15 cells infected with PCV2 and extracted cytoplasmic and nuclear proteins to determine whether G3BP1 influenced the localization of Cap and Rep proteins in PCV2 during the later stages of infection. The results in Figure 7G show that the overexpression of G3BP1 significantly promoted the accumulation of PCV2 Cap and Rep proteins in the cytoplasm 48 h after infection, suggesting that G3BP1 may be involved in the nuclear transfer of mature viruses.

## 3. Materials and Methods

### 3.1. Cells, Virus Strains, Plasmids, and Antibodies

PK15 cells, HEK293T cells, Hela cells, and PCV2-GD strain (GenBank: AY613854.1) were preserved at the Laboratory of Pig Disease Prevention and Control, National Engineering Research Center for Pig Breeding. Plasmids pCDNA3.1-3 × FLAG, pCMV-HA, IFNβ-Luc, and TK-Luc were maintained in our laboratory. NF-κB, IFNβ, and TK dual fluorescence reporter gene plasmids were generously provided by Professor Jun Cui of Sun Yat-sen University. Rabbit monoclonal antibodies against HA, mouse monoclonal antibodies against FLAG, rabbit monoclonal antibodies against histones, GAPDH, and actin, 488-labeled goat anti-rabbit and goat anti-mouse fluorescent monoclonal antibodies, and 594-labeled goat anti-rabbit and goat anti-mouse fluorescent monoclonal antibodies were purchased from Sigma Aldrich Corporation (St. Louis, MO, USA). Horseradish peroxidase-conjugated goat anti-mouse/anti-rabbit IgG (H + L) was purchased from Elabscience (Wuhan, China). Rabbit polyclonal antibodies against PCV2 Cap protein and PCV2 Rep protein were purchased from GeneTex (Southern California, USA). Mouse monoclonal antibodies against G3BP1 protein were procured from Proteintech (Wuhan, China).

### 3.2. Cell Culture and Virus Inoculation

PK-15 cells were cultured in high-glucose DMEM supplemented with 10% FBS in a sterile environment at 37 °C with 5% CO_2_. When cells reached approximately 90% confluency, they were passaged at a ratio of 1:4. The culture medium was first aspirated, cells were rinsed twice with sterile PBS, and then cells were treated with 0.25% trypsin at 37 °C for 3 min until they began to detach. Trypsin was removed, cells were gently tapped to dislodge them, fresh medium was added to neutralize digestion, and cells were resuspended and transferred into new culture vessels with the aforementioned medium. Excess cells were used for subsequent experiments. For viral infection experiments, cells at approximately 80% confluency were infected with PCV2 virus suspension prepared in high-glucose DMEM supplemented with 2% FBS to achieve an MOI = 1, ensuring the complete coverage of cells. Incubation was carried out at 37 °C with 5% CO_2_ for 2 h. After aspiration of the viral inoculum, cells were washed twice with sterile PBS, and fresh high-glucose DMEM with 2% FBS was added for continue incubation.

### 3.3. Cell Transfection and Gene Silencing Assay

We performed cell transfection experiments according to the Lipofectamine 3000 Transfection Reagent manual. Seed PK-15 cells were placed evenly into cell culture plates and allowed to grow to 70–90% confluency before transfection. Dilute Lipofectamine^TM^ 3000 Reagent diluted in Opti-MEMTM Medium was vortexed gently to mix. Dilute plasmid DNA in Opti-MEMTM Medium was mixed thoroughly, and then added to P3000TM Reagent and vortexed quickly to mix. The prepared reagents were mixed at a 1:1 ratio, left to stand for 15 min, and then evenly add to the cells, dropwise. For RNA interference experiments, we strictly followed the Lipofectamine^TM^ RNAi MAX Reagent manual. The detailed interference steps were as follows: PK-15 cells were seeded evenly into cell culture plates and allowed to grow to 60–80% confluency before transfection. Lipofectamine^TM^ RNAi MAX Reagent was diluted in Opti-MEMTM Medium and vortexed gently to mix them. SiRNA was diluted in Opti-MEMTM Medium and mixed thoroughly. The prepared reagents were mixed at a 1:1 ratio, left to stand for 5 min, and then evenly added to the cells, dropwise.

### 3.4. Real-Time Fluorescence Quantitative PCR Detection

Total RNA was extracted from the cell samples, following the instructions provided in the Promega GoScriptTM Reverse Transcription Mix manual, and then stored frozen for future use. Primers were designed using Primer 4.0 bioinformatics software based on gene sequences uploaded to GenBank. The primer sequences and GenBank accession numbers are listed in Table 1. Real-time fluorescent quantitative PCR detection was prepared using the Promega Eastep qPCR Master Mix (2X) dye-based fluorescence quantitative assay kit. We utilized a real-time fluorescent quantitative PCR instrument from Applied Biosystems, with the following reaction program: 95 °C for 2 min, followed by 40 cycles of 95 °C for 15 s and 60 °C for 30 s. Subsequently, we analyzed and compared the results.

### 3.5. Western Blotting

We prepared protein samples and loaded them onto a polyacrylamide gel for electrophoresis at 75 V for 20 min, and then increased this to 110 V until electrophoresis was complete. Wet transfer was conducted on ice at 110 V for 2 h to transfer proteins onto a PVDF membrane using cold transfer buffer. The membrane was blocked with skim milk blocking solution on a shaker at 80 rpm for 1 h. It was incubated with appropriately diluted primary antibody solution for 1 h or overnight at low temperature on a shaker at 80 rpm. The membrane was washed 5 times, for 5 min each time, with PBST on a shaker at 80 rpm. It was incubated with appropriately diluted secondary antibody solution for 1 h on a shaker at 80 rpm. Then, the membrane was washed 5 times, for 5 min each time, with PBST on a shaker at 80 rpm. Finally, the images were taken on a gel imaging system after soaking the membrane in luminescent solution.

### 3.6. Indirect Immunofluorescence Assay

The cells to be treated were fixed with 4% paraformaldehyde for at least 30 min, and then the cells were washed with PBS and closed with 1% BSA blocking solution at 37 °C for 1 h. The cells were penetrated with 0.2%Triton 100 for 15 min, washed with PBS three times, and then incubated with a primary antibody solution at 80 rpm for 1 h or overnight at a lower temperature. The cells were washed three times with PBST (PBS + Tween 20). Under dark conditions, the cells were incubated with a fluorescent secondary antibody solution at 80 rpm for 1 h. The cells were washed with PBST three times. The cells were stained with DAPI solution at room temperature for 5 min. The cells were washed with PBST three times. Fluorescence was observed under a fluorescence microscope and images were captured.

### 3.7. Co-IP Assay

After discarding the cell culture medium, cells were washed three times with PBS. Subsequently, WB cell lysis buffer supplemented with 1% protease inhibitor was added in sufficient quantity to cover the cells. Cells were disrupted by repeated pipetting until fully detached, and the lysate was collected. The lysate was then centrifuged at 4 °C, 14,000 rpm for 10 min. To prepare for immunoprecipitation (IP), 20 μL of Protein G Sepharose was added to a 1.5 mL centrifuge tube, followed by 800 μL of pre-chilled PBS. The mixture was inverted several times and centrifuged briefly, discarding the supernatant. This washing step was repeated three times. After the final wash, 500 μL of PBS was added, along with 5 μL of IP antibody. The tube was incubated overnight on a cold shaker. Post-incubation, the tube was briefly centrifuged, and the supernatant was discarded, retaining the agarose beads bound with the antibody. The beads were then washed three times with PBS, followed by elution of the protein bound to the agarose beads using elution buffer. Subsequent steps involved protein immunoblotting experiments, followed by result analysis.

### 3.8. Double Fluorescent Reporter Gene Assay

PK-15 cells were evenly seeded into 24-well plates and grown to approximately 70% confluence for cell transfection. Each well was transfected with 80 ng of IFNβ-Luc plasmid and 30 ng of TK-Luc plasmid. After 36 h of transfection, cells were lysed thoroughly with cell lysis buffer, and the lysate was collected and centrifuged at 12,000 rpm for 2 min. The supernatant was collected for subsequent experiments. Firefly luciferase and Renilla luciferase assay reagents were prepared. Then, 20 μL of cell lysate supernatant was transferred to a blank plate, and 100 μL of firefly luciferase assay reagent was added. After thorough mixing for 5 min, luminescence was measured at the appropriate wavelength using a luminometer. Subsequently, 100 μL of Renilla luciferase assay reagent was added, mixed for 5 min, and then luminescence was measured at the appropriate wavelength using the luminometer. Data analysis was then performed.

## 4. Discussion

This study investigated the mutual regulation between PCV2 and host G3BP1 protein, along with its underlying mechanisms. Previous findings have indicated that PCV2 infection in PK-15 cells upregulates G3BP1 expression. The transfection of PCV2 Cap and Rep plasmids into PK-15 cells also significantly enhanced G3BP1 expression, suggesting that PCV2 regulates G3BP1 expression through its Cap and Rep proteins. The subsequent overexpression of G3BP1 gene in PK-15 cells notably promoted PCV2 virus replication. Conversely, interference with G3BP1 expression in PK-15 cells significantly inhibited PCV2 replication, highlighting the crucial role of G3BP1 in PCV2 virus replication. Studies have shown that G3BP1 protein in HEK293T cells was found to interact with cGAS, promoting cGAS recognition of viral DNA and activation of the type I interferon pathway [10]. Additionally, G3BP1 facilitates RIG-I recognition of Sev viral RNA and promotes IFNβ production [11]. Based on these observations, we hypothesized whether G3BP1 regulates IFNβ production through the cGAS/STING and RIG-I/MDA5 signaling pathways, thereby further promoting PCV2 replication. Subsequently, we validated the impact of G3BP1 overexpression on these pathways during PCV2 infection in PK-15 cells. The results demonstrate that G3BP1 promotes the expression of cGAS, STING, RIG-I, MDA5, and MAVS induced by PCV2, thereby further regulating IFNβ production and promoting PCV2 replication.

PCV2 is one of the smallest known viral genomes to date. In contrast to viruses such as African swine fever virus (ASFV) and porcine reproductive and respiratory syndrome virus (PRRSV), PCV2 possesses a compact genome of 1767/1768 bp. This necessitates the conservation of genetic resources for essential protein expression and maximizes the utilization of host genes and proteins, thereby facilitating complex immune evasion mechanisms. For instance, a unique mechanism employed by PCV2 involves exploiting inflammatory factors to enhance its own replication. Our experimental findings demonstrate that the synergy between IFNβ and PCV2 infection inhibits the expression of Cyc A, causing nearly all cells to arrest in the S phase of the cell cycle. PCV2 replication relies on the utilization of host DNA polymerase during the DNA synthesis phase for its rolling-circle replication, which typically occurs during the host cell’s S phase. By inducing a substantial population of cells to remain in the S phase, PCV2 gains ample time and optimal conditions for enzymes, substrates, and energy for its genome replication. Concurrently, cell growth arrest severely impacts the innate antiviral immune response triggered by IFNβ. Prolonged cell cycle arrest further activates programmed cell death pathways, such as autophagy and apoptosis, which prior research indicates to significantly promote PCV2 replication and release. This suggests a potential mechanism by which IFNβ enhances PCV2 replication. Moreover, in PCV2 challenge experiments and disease model construction, PCV2 infection alone rarely manifests significant clinical symptoms, and immunostimulants are necessary [20,21]. This may be due to limited replication in the absence of inflammatory factor stimulation, thereby failing to produce pronounced clinical symptoms. Additionally, PCV2 often presents more severe clinical symptoms when co-infected with other pathogens, potentially due to its suppression of IFNβ-mediated antiviral effects and its impact on the cell cycle.

Recent studies have demonstrated that G3BP1 can form liquid-phase bodies within cells. Liquid-phase bodies are dynamic structures composed of various biomolecules, characterized by their ability to dynamically aggregate and disperse dynamically, in contrast to the solid crystalline state or uniformly dispersed solution state [22,23]. They resemble droplets that can dynamically assemble and disassemble. The interaction between PCV2 Rep protein and Cap protein is critical for the replication and packaging processes of PCV2. During viral replication, Rep protein binds to specific initiation sites on the viral genome, thereby initiating the replication process. Once the replication of the viral genome is complete, the newly synthesized genome is packaged into viral particles by Rep protein and other auxiliary proteins. Cap protein is responsible for forming the viral capsid during this process, encapsulating the replicated genome to form mature viral particles. The interaction between Rep and Cap protein involves not only direct physical interactions but also complex regulatory networks. These include the participation of other auxiliary proteins and regulatory factors within the cellular environment, such as timing of the viral replication process and changes in intracellular conditions. In this context, the different domains of G3BP1 in this paper can interact with viral proteins Cap and Rep proteins. Moreover, G3BP1 can induce the formation of a liquid phase in the cytoplasm, which can form an independent space, which provides favorable conditions for the transport of the virus and escape from the innate immune response within the cell.

G3BP1 protein is one of the core proteins involved in the formation of stress granules. To date, there have been no reports linking PCV2 to the induction of stress granule formation. This study demonstrates that PCV2 infection and PCV2 Cap and Rep transfection into PK-15 cells induce an increase in G3BP1 expression. Additionally, PCV2 Cap interacts with the RGG domain of G3BP1, while PCV2 Rep interacts with both the NTF2 and RRM domains of G3BP1. The NTF2 domain of G3BP1 is associated with its nuclear transport and cellular localization, serving as a carrier that mediates oligomerization with other partner proteins [24]. The RGG domain of G3BP1 is crucial for recruiting the host’s translation machinery [25]. The RRM domain includes two short sequences, RNP1 and RNP2, which consist of conserved hydrophobic amino acids critical for RNA binding [26]. The interaction of PCV2 Cap and Rep with G3BP1 leads to the sequestration of cellular components and the subsequent formation of stress granules. Research indicates that PRRSV relies on protein kinase R-like endoplasmic reticulum kinase (PERK)-induced stress granule formation [27]. Stress granules are associated with the inhibition of virus replication complexes and host translation [14]. These formed stress granules encapsulate PRRSV genetic material, thereby inhibiting its replication. Additionally, porcine epidemic diarrhea virus (PEDV) induces stress granule formation but also induces caspase-8-mediated cleavage of G3BP1, which disrupts stress granules and promotes virus replication. Clinically, co-infections of PCV2 with PRRSV or PEDV are common and exhibit more severe symptoms compared to single infections [28,29,30,31]. This phenomenon can be explained by the mechanism whereby co-infections induce a significant production of inflammatory factors, including IFNβ, and synergistically affect the cell cycle, further promoting PCV2 replication. The increased replication of PCV2 results in elevated levels of Cap and Rep proteins, which interact with G3BP1, the key core protein induced by PEDV and PRRSV-induced stress granules, thereby inhibiting stress granule formation and further facilitating the replication of PEDV and PRRSV. This mechanism may also contribute to the immunosuppression caused by PCV2.

In summary, our study found that PCV2 infection and transfection of PCV2 Cap and Rep both promote the expression of G3BP1. G3BP1, in turn, enhances the expression of cGAS, STING, RIG-I, MDA5, and MAVS induced by PCV2, thereby facilitating the production of IFNβ. IFNβ collaborates with PCV2 infection to inhibit the expression of Cyc A, leading to cell cycle arrest predominantly in the S phase, which provides favorable conditions for virus replication. Furthermore, the RGG domain of G3BP1 interacts with PCV2 Cap, while its NTF2 and RRM domains interact with PCV2 Rep. These findings offer new insights into the pathogenesis and immune evasion mechanisms of PCV2.

## Figures and Tables

**Figure 1 ijms-26-01083-f001:**
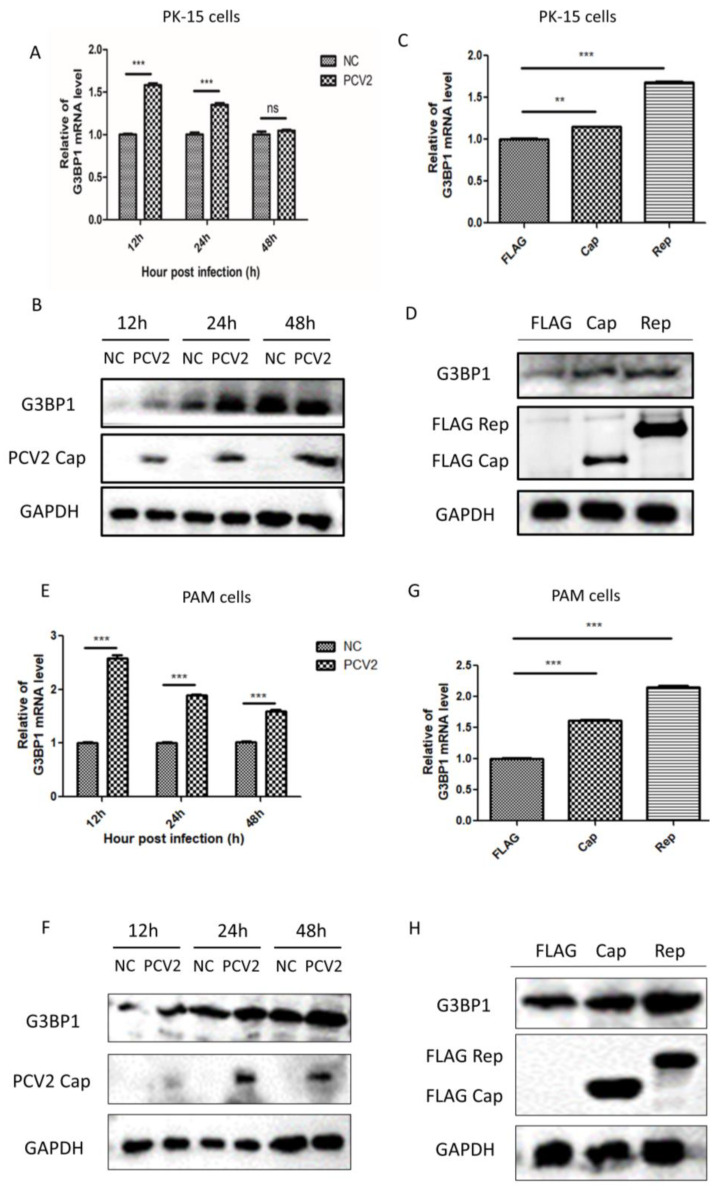
PCV2 infection significantly promotes the expression of G3BP1. (**A**,**B**) Infection of PK-15 cells with PCV2 increased both the mRNA and protein levels of G3BP1. (**C**,**D**) PCV2 Cap and Rep protein-transfected PK-15 cells promote the expression of G3BP1 mRNA and protein level. (**E**,**F**) PCV2 infection of PAM cells promoted the mRNA and protein expression of G3BP1. (**G**,**H**) PCV2 Cap and Rep protein transfected PAM cells to promote the expression of G3BP1 mRNA and protein level. ** *p* < 0.01. *** *p* < 0.001 “ns” means no statistical significance.

**Figure 2 ijms-26-01083-f002:**
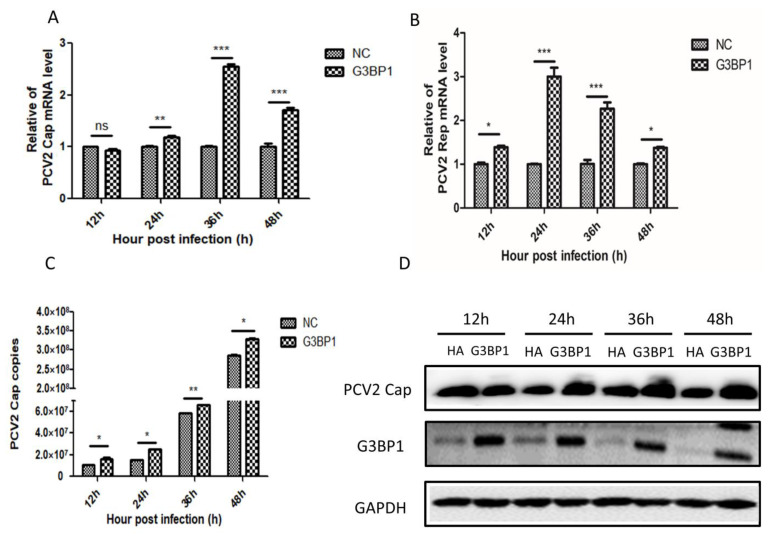
Overexpression of G3BP1 promotes PCV2 replication. (**A**,**B**) The overexpression of G3BP1 in PK15 cells promoted the expression of PCV2 Cap and Rep mRNA at different time points. (**C**) Effect of overexpression of G3BP1 on PCV2 gene copy number in PK15 cells. (**D**) Overexpression of G3BP1 in PK15 cells promoted the expression of PCV2 Cap protein. * *p* < 0.05, ** *p* < 0.01. *** *p* < 0.001 “ns” means no statistical significance.

**Figure 3 ijms-26-01083-f003:**
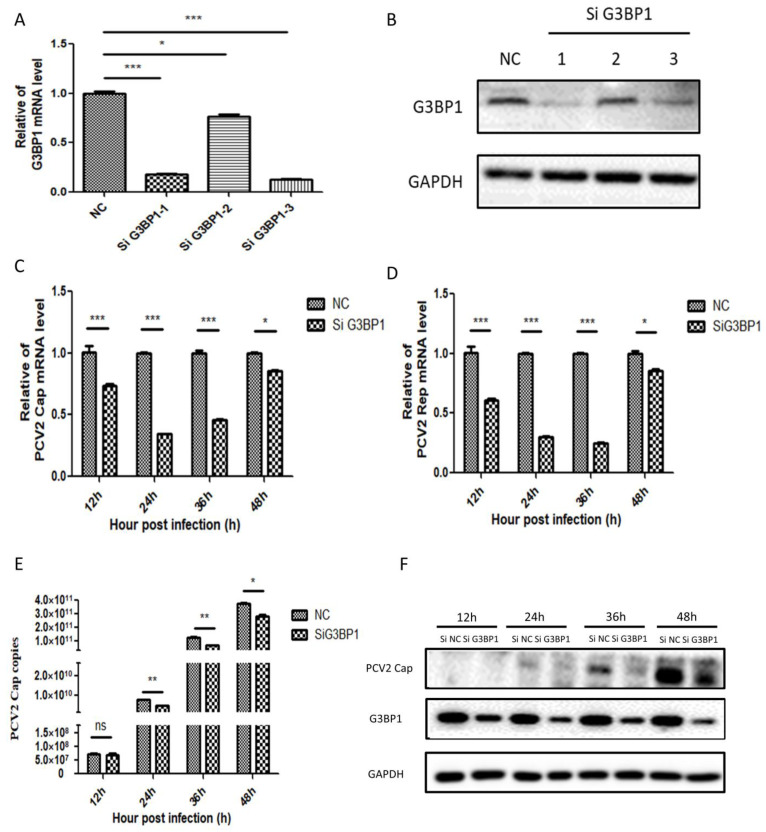
Silencing the expression of G3BP1 inhibited the replication of PCV2. (**A**,**B**) The silencing effect of G3BP1 after transfection of three SiRNAs into PK15 cells was determined from gene expression and protein levels. (**C**,**D**) Silencing G3BP1 expression in PK15 cells inhibited PCV2 Cap and Rep mRNA expression at different time points. (**E**) Effect of silencing G3BP1 expression on PCV2 virus copy number at different time points in PK15 cells. (**F**) Silencing G3BP1 inhibited the expression of PCV2 Cap protein in PK15 cells. * *p* < 0.05, ** *p* < 0.01. *** *p* < 0.001 “ns” means no statistical significance.

**Figure 4 ijms-26-01083-f004:**
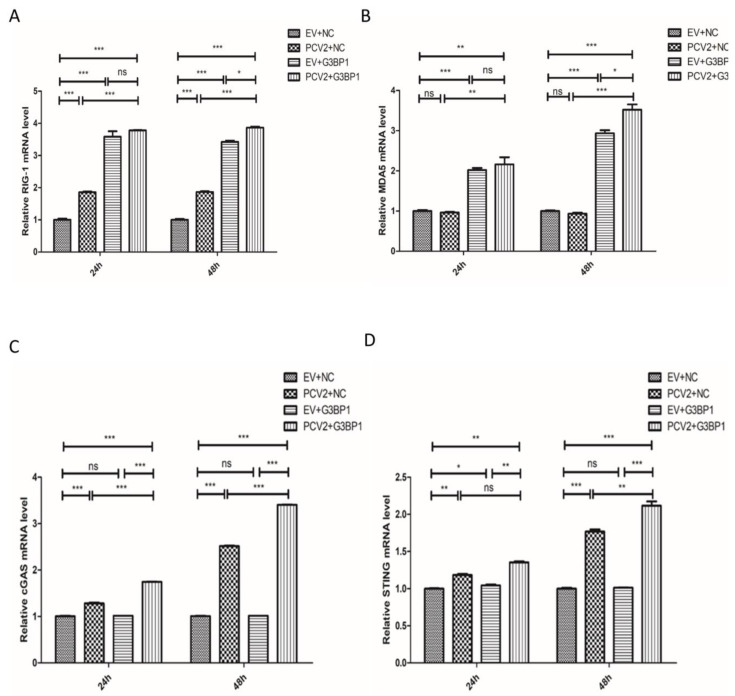
G3BP1 promotes PCV2-induced expression of RIG-I, MDA5, MAVS, cGAS, and STING. (**A**,**B**) Overexpression of G3BP1 in PK15 cells promotes PCV2-induced transcription of RIG-I and MDA5 mRNA levels. (**C**,**D**) Overexpression of G3BP1 in PK15 cells promotes the transcription of PCV2-induced cGAS and STING molecules. * *p* < 0.05, ** *p* < 0.01. *** *p* < 0.001 “ns” means no statistical significance.

**Figure 5 ijms-26-01083-f005:**
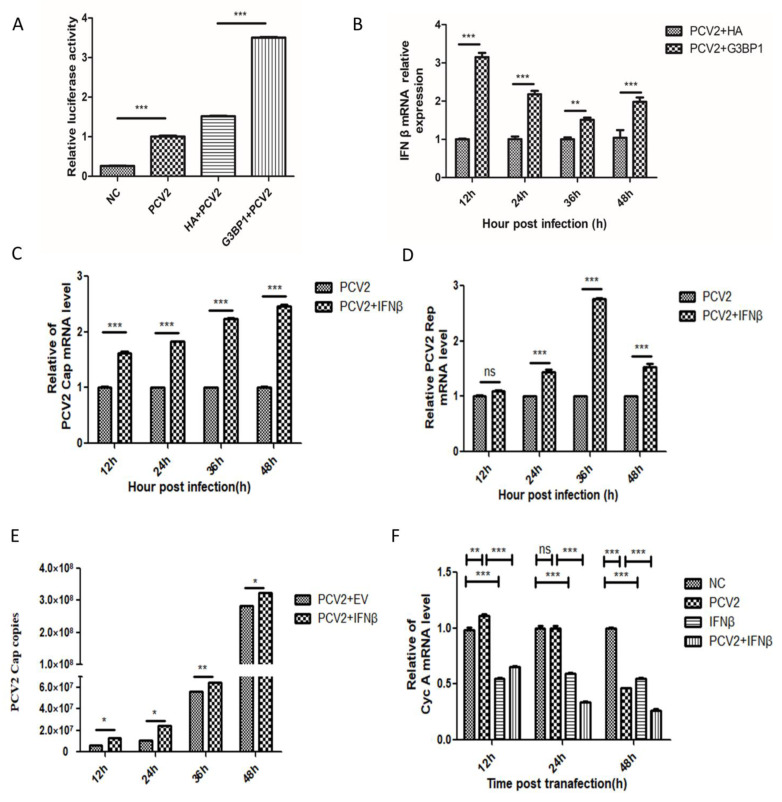
G3BP1 promotes PCV2 replication by triggering IFNβ expression and arresting regulating cell cycle at S stage. (**A**) Double fluorescence reporter gene detection of G3BP1 overexpression promotes PCV2-induced IFN-β promoter transcription. (**B**) Overexpression of G3BP1 in PK15 cells promoted PCV2-induced IFNβ mRNA levels at different time points. (**C**–**E**) Exogenous IFNβ of 500 ng/mL stimulated PK-15 cells promote PCV2 Cap and Rep mRNA transcription and increase viral copy number. (**F**) Effects of PCV2 infection and 500 ng/mL exogenous IFN-β stimulation on the expression of Cyc A mRNA in PK-15 cells. * *p* < 0.05, ** *p* < 0.01. *** *p* < 0.001 “ns” means no statistical significance.

**Figure 6 ijms-26-01083-f006:**
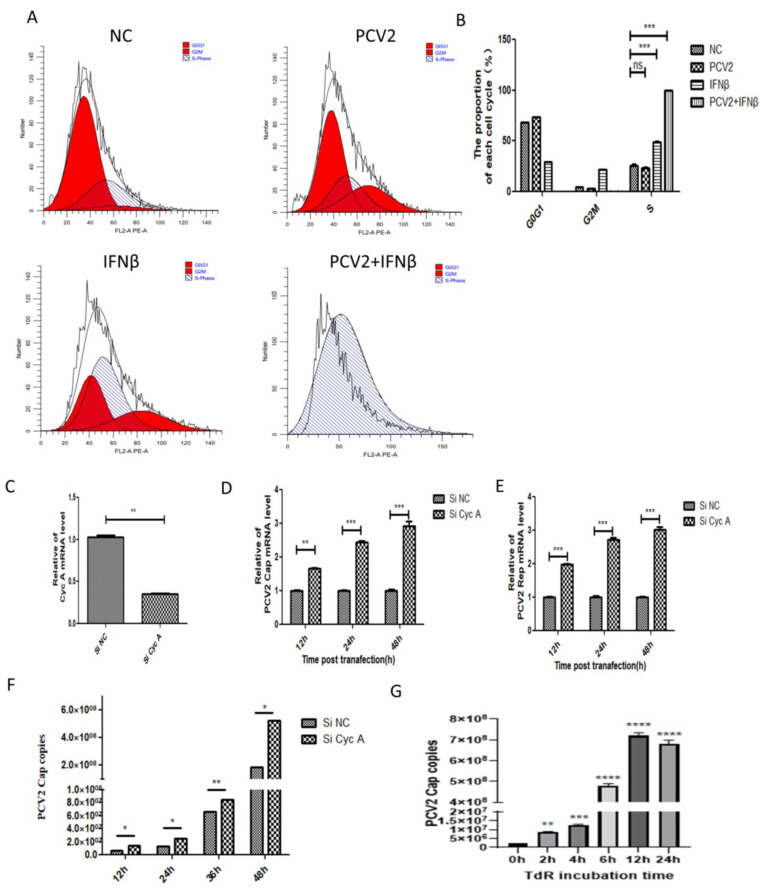
G3BP1 promotes PCV2 replication by triggering IFNβ expression and arresting regulating cell cycle at S stage. (**A**,**B**) The effect of PCV2 infection and 500 ng/mL exogenous IFN-β stimulation on the cell cycle of PK-15 cells was detected by flow cytometry. (**C**) SiRNA silencing efficiency of Cyc A. (**D**–**F**) Silencing the expression of Cyc A promoted the transcription of PCV2 Cap and Rep mRNA and the increase in viral copy number. (**G**) PCV2-infected PK-15 cells were treated with TdR for different time lengths to observe the effect on PCV2 virus copy number. * *p* < 0.05, ** *p* < 0.01. *** *p* < 0.001 “ns” means no statistical significance.

**Figure 7 ijms-26-01083-f007:**
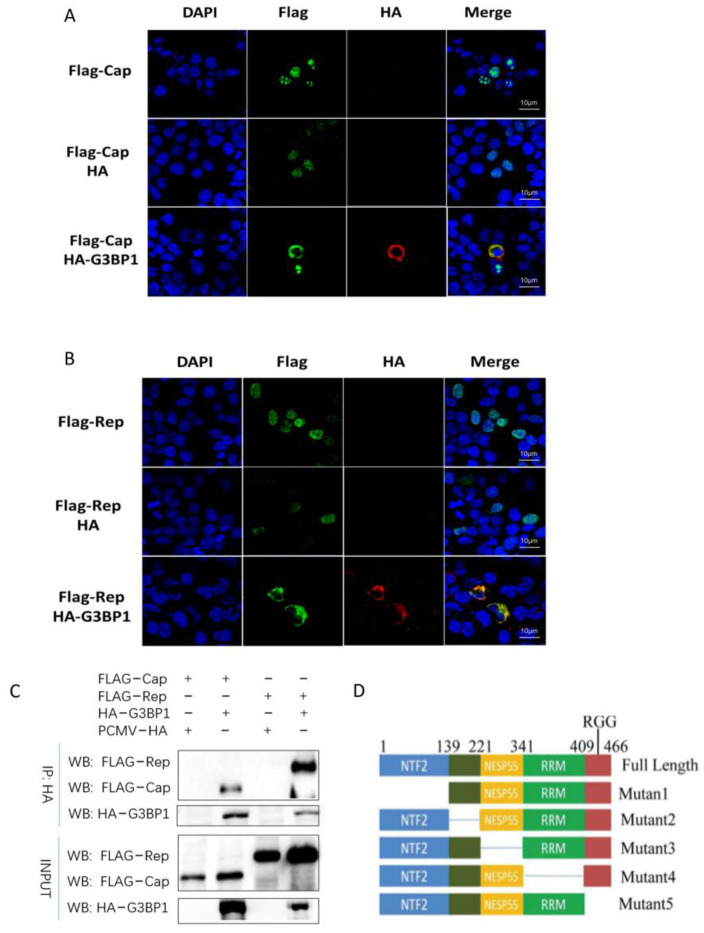
PCV2 Cap and Rep proteins interact with G3BP1. (**A**,**B**) The co-localization of Cap and G3BP1 and Rep and G3BP1 were observed by confocal laser. The green fluorescence represents the Flag−Cap and Flag−Rep proteins, and the red fluorescence represents the HA-G3BP1 protein. (**C**) CO-IP verified that G3BP1 interacts with PCV2 Cap and Rep proteins. (**D**) Schematic diagram of deletion of 5-segment domain of G3BP1 gene. (**E**) PCV2 Cap interacts with the RGG functional region of G3BP1. (**F**) PCV2 Rep interacts with NTF2 functional region and RRM functional region of G3BP1. (**G**) Nucleolar and cytoplasmic localization of PCV2 Cap and Rep proteins by overexpression of G3BP1. * *p* < 0.05, ** *p* < 0.01. *** *p* < 0.001 “ns” means no statistical significance.

**Table 1 ijms-26-01083-t001:** qRT-PCR primer sequence.

Primer Name	Primer Sequence (5′-3′)	Genbank Serial Number
Sus G3BP1 F	aggtgaggtccgtctgaatg	NM_001205405.1
Sus G3BP1 R	cccttcccactccaaatcct	
PCV2 Cap F	gagcagggccagaattcaac	MN653212.1
PCV2 Cap R	aaaaccacagtcagaacgcc	
PCV2 Rep F	actgctgtgagtaccttgct	DQ346683.1
PCV2 Rep R	ggtttccgggtctgcaaaat	
Sus IFNβF	tccaccacagctctttccat	NM_001003923.1
Sus IFNβR	ctggaattgtggtggttgca	
Sus RIG-I F	gcagacatgggatgaagcag	NM_213804.2
Sus RIG-I R	agcatctccaaccacagtga	
Sus MDA5 F	cttcagcctacagtggtgga	MF358967.1
Sus MDA5 R	atcggtgcctgttaattcgc	
Sus cGAS F	ggaagttccccgaattcagc	XM_013985148.2
Sus cGAS R	ttcctctccacagtgacacc	
Sus STING F	actcccttgcagaccttgtt	FJ455509.1
Sus STING R	cctctgtgggttcctggtag	
Sus MAVS F	cttacctgtcctgcctcaca	MK302496.1
Sus MAVS R	ctggtagacacgggacactt	
Sus actin F	ccgagatctcaccgactacc	EU655628.1
Sus actin R	ctcgtagctcttctccaggg	

## Data Availability

No new data were created.

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
