# Peer review of "G3BP1 Regulates the Cell Cycle by Promoting IFNβ Production to Promote PCV2 Replication and Promotes Nuclear Transfer of Viral Proteins by Direct Binding"

_ijms, 2025, doi:10.3390/ijms26031083_

Round 1

Reviewer 1 Report

Comments and Suggestions for Authors

Zhang et al. here reports that G3BP1 regulates the cell cycle by promoting IFN-beta production to promote porcine circovirus type 2 (PCV2) replication. This study facilitates to better understanding the pathogenesis and immune evasion mechanism of PCV2 infection. However, there are many concerns to be addressed by the reviewer as listed in comments for the authors.

1. Supporting data [Title "......viral assembly"] is not available in the study, the title of the manuscript needs to be modified.

2. P24, "potentially aiding virus assembly in cytoplasm", PCV2 virus assembly occurs in the cytoplasm?

3. P26-27, the sentence needs to be rewritten.

4. P48-50, "......between its own proteins and nucleic acids with various host cell factors" is not clear.

5. P59-63, these two sentences are unclear and illogical.

6. P150 "Western blot" should be "Western blotting".

7. P212-214, "This translation maintains clarity and precision......" is not clear, what does it mean?

8. Figure 3, A, B, D, and E data are not strongly related to this study, do deletion or make a modification?

9. Many redundant, unclear, or incorrect descriptions need to be modified.

Comments on the Quality of English Language

The English could be improved such as more concise, less grammatical errors, and so on.

Author Response

Dear reviewer,

  Thank you very much for your comments and professional advice. These opinions help to improve academic rigor of our article. Based on your suggestion and request, we have made corrected modifications on the revised manuscript. Meanwhile, the language has also been refined, and redundant, unclear, or incorrect descriptions have been corrected and marked in the text. In addition, we have resubmitted a new manuscript in the revised state, with the revisions highlighted in red.

  Furthermore, we would like to show the details as follows:

Comments 1: [ Supporting data [Title "......viral assembly"] is not available in the study, the title of the manuscript needs to be modified.]

Response 1: [Thank you for pointing this out. We agree with this comment. We really lacked sufficient evidence to prove that G3BP1 in combination with viral Cap and Rep promotes viral assembly, so we changed the title to" G3BP1 regulates the cell cycle by promoting IFNβ production to promote PCV2 replication and directly binds viral proteins to promote viral nuclear transfer”, and we have also made revisions in the text description.]

Comments 2: [ P24, "potentially aiding virus assembly in cytoplasm", PCV2 virus assembly occurs in the cytoplasm?]

Response 2: [Thank you for pointing that out. We are aware of the error in our presentation and have made changes in the text. Studies have suggested that: PCV2's only nucleocapsid protein, Cap protein, has a nuclear localization signal and can actively enter the nucleus, so PCV2 is assembled in the nucleus and exits the nucleus with the help of P32 protein. The references are as follows: Cellular p32 Is a Critical Regulator of Porcine Circovirus Type 2 Nuclear Egress. DOI: 10.1128/JVI.00979-19.]

Comments 3: [P26-27, the sentence needs to be rewritten.]

Response 3: [Thank you for pointing that out. “This study provides novel insights into the interplay between G3BP1 and DNA viruses, offering a reference for further exploration into PCV2 pathogenesis and immune evasion mechanisms.”We have rewritten this sentence as  “This research provides a foundation for further investigation into the immune evasion mechanisms of PCV2. ”]

Comments 4: [P48-50, "......between its own proteins and nucleic acids with various host cell factors" is not clear.]

Response 4: [Thank you for pointing that out. Because it is not clearly expressed, we have made additional explanations in the manuscript. The supplementary content is “PCV2 cap binds to DEAD-box RNA helicase 21 to promote viral replication (Zhou et al., 2024). In addition, PCV2 Cap inhibits PKR activation by interacting with Hsp40 (Lv et al.,2021). ” The supplementary content is intended to explain that PCV2 achieves host immune escape mainly through the interaction between its own protein and a variety of host cytokines. ]

Comments 5: [ P59-63, these two sentences are unclear and illogical.]

Response 5: [Thank you for pointing that out. Since the two sentences you pointed out are unclear and illogical, we have rewritten this paragraph as“It is worth noting that PCV2 infection increases the protein levels of phosphorylated IRF3 (p-IRF3), mitochondrial antiviral signaling protein (MAVS), retinoic acid-inducible gene I (RIG-I), and melanoma differentiation-associated protein 5 (MDA-5)(Huang et al., 2017). The activation of these host innate immune molecules further promotes the expression of downstream interferon-stimulated genes (ISGs). This dysregulates intracellular inflammatory cytokine expression and disrupts cellular homeostasis, creating favorable conditions for the invasion of other diseases (Ramamoorthy and Meng, 2009). Interestingly, while type I interferons (IFN-α and IFN-β) and interleukin-2 (IL-2) are well-known for their antiviral effects, their stimulation paradoxically enhances the replication of PCV2 (Huang et al., 2017; Huang et al., 2018).” ]

Comments 6: [ "Western blot" should be "Western blotting".]

Response 6: [Thank you for pointing that out. we have revised it in the paper]

Comments 7: [ P212-214, "This translation maintains clarity and precision......" is not clear, what does it mean?]

Response 7: [Thank you for pointing that out. "This translation maintains clarity and precision......". This sentence is a remark in the manuscript of the article. Due to our carelessness, we did not delete it after the manuscript was completed. For this, we are very sorry. In this article, we have made significant revisions to this paragraph and highlighted it in red font.]

Comments 8: [ Figure 3, A, B, D, and E data are not strongly related to this study, do deletion or make a modification?]

Response 8: [Thank you for pointing that out. Figure 3, A, B, D, and E data are not strongly related to this study. We agree with your point of view, we have deleted in the article and revised the content in the article.]

Comments 9: [ Many redundant, unclear, or incorrect descriptions need to be modified.]

Response 9: [Thank you for pointing that out. According to your reminder, we have made many changes to the descriptions in the article after careful study. Include unclear and incorrect descriptions that you indicate and do not indicate, and are marked in red in the text.]

Comments 10: [ The English could be improved such as more concise, less grammatical errors, and so on.]

Response 10:[ We apologize for the poor language of our manuscript. We worked on the manuscript for a longtime and the repeated addition and removal of sentences and sections obviously led to poor readability. We have now worked on both language and readability and have also involved native English speakers for language corrections. ]

 We would like to thank you for your professional review work, constructive comments, and valuable suggestions again on our manuscript. If there are any incorrect answers or questions in the manuscript, please do not hesitate to let us know. Looking forward to hearing from you.

Yours sincerely,

Changxu Song, Ph.D., Professor,

College of Animal Science, South China Agricultural University,

Director of Swine Disease Prevention and Control Laboratory, National Engineering Technology Research Center of Breeding Pig Industry

cxsong@scau.edu.cn/cxsong2004@163.com

Reviewer 2 Report

Comments and Suggestions for Authors

This paper from Zhang et al., studies the interaction between PCV2 and the host protein G3BP1, showing that PCV2 manipulates G3BP1 to enhance its replication by inducing IFN production and arresting the host cell cycle in the S phase. Also, the research demonstrates that PCV2’s Cap and Rep proteins directly bind specific G3BP1 domains to facilitate viral assembly, shedding light on the PCV2’s pathogenesis and immune evasion strategies.

Overall, the study is well conducted with good methodological rigor and results are mostly supported by data. In particular, validation of the results with co-immunoprecipitation, fluorescence microscopy, and gene silencing, are well conducted to support the findings.
However, some points of the paper can be improved and may need to be addressed to strengthen the conclusions. Using PK-15 cells, an immortalised cell line raises concerns about physiological relevance and how these cells recapitulate the infection in in vivo conditions. Can the author comment on this or perform a few experiments (at least the key ones) in another model cell line? Additionally, the authors should expand more in the discussion on the relationship and interactions between G3BP1 and viral proteins during infection.
Furthermore, the authors mentioned that cell cycle arrest is implicated in facilitating PCV2 replication, however, the direct relationship between this arrest and viral replication efficiency was not properly dissected. The author may need to perform an experiment using synchronised cell cycle models.
The authors may also perform a co-infection experiment to validate its proposed mechanism in more real-world conditions and add relevance to the study.
Overall, I recommend accepting this paper for publication after minor revisions.

Minor points:
Figure 1F – include quantification of western blot
Figure 2F – G3BP1 bands are mostly saturated – replace with less exposed western blot
Figure 5F – bands are saturated, please replace them with less exposed western blot

Author Response

Dear reviewer,

  Thank you for your decision and constructive comments on my manuscript. All of us authors have carefully read the comments that you have given us, and have discussed and revised each of these issues. In addition, we have resubmitted a new manuscript in the revised state, with the revisions highlighted in blue. The following is my list of revisions.

Comments 1: [Using PK-15 cells, an immortalised cell line raises concerns about physiological relevance and how these cells recapitulate the infection in in vivo conditions. Can the author comment on this or perform a few experiments (at least the key ones) in another model cell line?]

Response 1: [Thank you for pointing this out. Your guidance is very important to the improvement of our article. Indeed, the use of a immortalised cell line does not indicate the prevalence of G3BP1 in PCV2 infection. In order to solve this problem, we added some data. As shown in Figure 1E-H, we conducted PCV2 infection and protein transfection experiments with primary porcine alveolar macrophages (PAM), and the phenomenon was consistent with that of PK15 cells, confirming once again the up-regulation of G3BP1 expression during PCV2 infection. This implies that upregulation of G3BP1 is universal during PCV2 infection of host cells. The revision of this part of the content is marked in blue font in the article.]

Comments 2: [Additionally, the authors should expand more in the discussion on the relationship and interactions between G3BP1 and viral proteins during infection.]

Response 2: [Thank you for pointing this out. In the discussion section of this paper, the relationship and interaction between G3BP1 and viral proteins during infection were discussed in depth. We suggest that the different domains of G3BP1 in this paper can interact with the viral proteins Cap and Rep proteins. Moreover, G3BP1 can induce the formation of a liquid phase in the cytoplasm, forming an independent space that provides favorable conditions for the virus to transport within the cell and evade the innate immune response.]

Comments 3: [Furthermore, the authors mentioned that cell cycle arrest is implicated in facilitating PCV2 replication, however, the direct relationship between this arrest and viral replication efficiency was not properly dissected. The author may need to perform an experiment using synchronised cell cycle models. The authors may also perform a co-infection experiment to validate its proposed mechanism in more real-world conditions and add relevance to the study.]

Response 3: [Thank you for pointing this out. Your guidance is very important to the improvement of our article. To further verify whether the increased PCV2 replication is related to the duration of cell cycle residence in the S phase, we treated PK-15 cells for 2, 4, 6, 12, and 24 hours with the reversible DNA synthesis inhibitor thymidine riboside (TdR) to prolong the duration of cell residence in the S phase. Concurrently, PCV2 virus with MOI=1 was inoculated and the difference in virus copy number was observed after a 24 hour culture period. The re-sults were shown in Figure 6G. With the increasing duration of TdR treatment time of PK-15 cells, the virus copy number increased significantly. This indicates that the en-hancement of PCV2 replication is closely related to the length of time the cell remains in the S phase of the cell cycle. The revision of this part of the content is marked in blue font in the article. Your suggestions have opened new ideas for us and provided new insights for the expansion of PCV2 virus. Thank you again for your professional suggestions.]

Comments 4: [Minor points: Figure 1F – include quantification of western blot.
Figure 2F – G3BP1 bands are mostly saturated – replace with less exposed western blot. Figure 5F – bands are saturated, please replace them with less exposed western blot]

Response 4: [Thank you for pointing this out. For the unqualified pictures you pointed out, we have corrected and replaced the data.]

We would like to thank you for your professional review work, constructive comments, and valuable suggestions again on our manuscript. If there are any incorrect answers or questions in the manuscript, please do not hesitate to let us know. Looking forward to hearing from you.

Yours sincerely,

Changxu Song, Ph.D., Professor,

College of Animal Science, South China Agricultural University,

Director of Swine Disease Prevention and Control Laboratory, National Engineering Technology Research Center of Breeding Pig Industry

cxsong@scau.edu.cn/cxsong2004@163.com

Round 2

Reviewer 1 Report

Comments and Suggestions for Authors

No further comments.